# High Expression of a tRNA^Pro^ Derivative Associates with Poor Survival and Independently Predicts Colorectal Cancer Recurrence

**DOI:** 10.3390/biomedicines10051120

**Published:** 2022-05-12

**Authors:** Panagiotis Tsiakanikas, Panagiotis G. Adamopoulos, Dimitra Tsirba, Pinelopi I. Artemaki, Iordanis N. Papadopoulos, Christos K. Kontos, Andreas Scorilas

**Affiliations:** 1Department of Biochemistry and Molecular Biology, Faculty of Biology, National and Kapodistrian University of Athens, 15701 Athens, Greece; ptsiak@biol.uoa.gr (P.T.); padamopoulos@biol.uoa.gr (P.G.A.); dtsirmpa@biol.uoa.gr (D.T.); partemaki@biol.uoa.gr (P.I.A.); 2Fourth Surgery Department, National and Kapodistrian University of Athens, University General Hospital “Attikon”, 12462 Athens, Greece; ipapado@med.uoa.gr

**Keywords:** tRNA-derived small RNA (tsRNA), tRNA-derived fragment (tRF), tRNA half, molecular tumor biomarkers, prognosis, colon carcinoma, colon cancer progression, survival, real-time quantitative PCR (qPCR), functional in silico analysis

## Abstract

Colorectal cancer (CRC) is the second most lethal cause of cancer-related deaths in Europe. Fragments of tRNA^Pro^ are conserved among vertebrates, characterized by pleiotropic regulatory functions and have been found to discriminate colorectal tumors from normal colorectal mucosa. In the current study, we investigated the prognostic utility of 5′-tiRNA-Pro^TGG^ levels in CRC. For this purpose, total RNA was extracted from 155 malignant colorectal tumors and 74 adjacent non-cancerous tissue specimens, polyadenylated and reverse-transcribed using an oligo-dT adapter as primer. Real-time quantitative PCR (qPCR) was used to assess the levels of 5′-tiRNA-Pro^TGG^. Kaplan-Meier survival analysis demonstrated that high 5′-tiRNA-Pro^TGG^ levels predict both poor disease-free survival (DFS) and overall survival (OS) of CRC patients. Of note, high 5′-tiRNA-Pro^TGG^ levels retain their unfavorable prognostic value in patients with rectal cancer and/or moderately differentiated CRC (grade II). More importantly, multivariate cox regression analysis highlighted that the overexpression of 5′-tiRNA-Pro^TGG^ constitutes an adverse prognostic factor predicting short-term relapse of CRC patients independently of the established prognosticators in CRC. Finally, bioinformatics analysis unveiled a potentially critical role of 5′-tiRNA-Pro^TGG^ regarding the maintenance of cellular homeostasis, signaling, cell communication, and cellular motility.

## 1. Introduction

According to global cancer statistics, as presented by the Global Cancer Observatory (GCO) in 2018, colorectal cancer (CRC) accounts for approximately 12% of annually diagnosed cancers, being reported as the second most common cause of cancer-related deaths in Europe. The progression of benign colorectal incidentalomas to malignant neoplasms is a lengthy process that is mediated through genomic instability, aberrant DNA methylation and mutations in crucial oncogenes and/or tumor-suppressor genes [1]. Early detection of CRC remains a prerequisite for the effective management and/or treatment of the disease. As a result, the current CRC prevention strategies are mainly focused on the detection and subsequent removal of colonic adenomas as well as the pre-symptomatic diagnosis of early CRC. Implementation of such strategies can effectively attenuate CRC incidence rates and mortality [2,3]. However, sporadic CRC is characterized by a genomic complexity and heterogeneity, which can directly affect the clinical behavior of the tumor [4]. Molecular biomarkers are used for efficient screening and personalized management of CRC patients. The research on multipurpose colorectal cancer biomarkers is an emerging point of interest, guided by the ever-increasing knowledge of the molecular background of CRC. Nevertheless, the identification of molecular tumor markers that could be used for personalized treatment remains a challenge [5,6].

The tumor-node-metastasis (TNM) classification system for colorectal tumors constitutes the most robust prognosticator to determine the survival of patients. Even in the later revisions of the TNM staging, the extent of the tumor (T), the nodal positivity (N) and the presence or absence of metastatic disease (M) retain their strong prognostic potential regarding the outcome of the disease and guide the clinical management of the patients [7]. As we are moving away from a one-size-fits-all model to personalized therapeutic approach, the upcoming necessity to identify novel molecular biomarkers becomes even stronger.

Towards that direction, the pleiotropic action of non-coding RNAs (ncRNAs) in the regulation of gene expression makes them as putative diagnostic, prognostic or predictive molecular markers not only in CRC, but also in a wider spectrum of human diseases [8,9,10,11]. Small ncRNAs (sncRNAs) constitute a class of ncRNAs, which are characterized by a length shorter than 200 nucleotides (nt) and are heavily implicated in the regulation of gene expression. Among these small ncRNAs, an emerging frontier is represented by microRNAs (miRNAs) that have obtained a clinical relevance only after much research effort [12,13,14,15].

Transfer RNA (tRNA)-derivatives, which are widely known as tRNA-derived small RNAs (tsRNAs), are endogenous single-stranded sncRNAs with an approximate length of 14–40 nt. There are mainly two types of tsRNAs, including tRNA-derived stress-induced RNAs (tiRNAs) and tRNA-derived fragments (tRFs), which differ in the cleavage position of the precursor or mature tRNA transcripts, as well as in their length [16]. In brief, tiRNAs are produced through specific angiogenin (ANG)-dependent endonucleolytic cleavage of mature tRNAs inside their anticodon loop. Based on the cleavage site, they can be categorized in two groups, 5′- and 3′-tiRNAs [17]. In addition to tiRNAs, tRFs can be discriminated in three additional classes, namely tRF-5, tRF-3 and i-tRF, which represent shorter fragments (14–30 nt) and are mapped to either 5′-, 3′-ends or entirely internal sites of the mature tRNA, respectively. Finally, tRF-1 is the most variable class of tRFs, derived by endonucleolytic cleavage of 3′-trailer sequences of the precursor tRNA [18].

Recent scientific evidence has shown that tsRNAs participate in the regulation of the expression of protein-coding genes and as a result they are implicated in a wide range of molecular mechanisms underlying cancer development by possessing several biological roles. More specific, tsRNAs can regulate the expression of cancer-related genes, by forming complexes with Argonaute (AGO) and PIWI family members, imitating the function principles of miRNAs and PIWI-interacting RNAs (piRNAs) [19,20]. Additionally, they can bind to RNA binding proteins (RBPs), leading to the post-transcriptional destabilization or translational repression of several oncogenic transcripts and acting as tumor-suppressors [21].

tsRNAs are versatile RNA molecules, playing a crucial role in the processes of malignant transformation, since they are implicated in the regulation of ribosomal biogenesis, translational efficiency, cell cycle, cell proliferation, migration and apoptosis [16,17,22]. Although the biological role of tRNA derivatives has been neglected, emerging evidence in the literature place them among the key regulators of malignant transformation in CRC. A tRF-3 derived from tRNA^Leu^, previously known as miR-1280, has been reported to be deregulated in colorectal tumors, repressing the Notch pathway [23]. Furthermore, tiRNA^TyrGTA^ is associated with apoptotic process of colonic epithelial cells and peroxisome proliferator-activated receptor (PPAR) signaling pathway [24]. Finally, several tsRNAs contribute to the progression of CRC by regulating cGMP-PKG molecular pathway and vitamin metabolism, whereas fragments derived from precursor tRNAs can discriminate between colorectal adenomas and adenocarcinomas [25,26].

A variety of well-conserved tRNA^Pro^ fragments have been found to modulate gene expression in different developmental stages across vertebrates. Regarding the present study, two main reasons prompted us to evaluate the expression levels of 5′-tiRNA-Pro^TGG^, as a novel prognostic marker in CRC patients. First, these fragments have been reported to be conserved in humans and more interestingly to be differentially expressed between CRC tumor samples and adjacent colorectal mucosa [27]. Second, OncotRF database, which exploits datasets from The Cancer Genome Atlas Program (TCGA) to unveil putative tRFs as prognosticators in a plethora of human malignancies, provides evidence regarding the clinical value of several tRNA^Pro^ derivatives. Specifically, overexpression of tRNA^Pro^ fragments is significantly associated with worse disease-free survival (DFS) and overall survival (OS) in breast, stomach, prostate, and head and neck cancer as well as in melanoma [28]. However, the available databases offer limited information regarding the potential clinical value of tRNA^Pro^ fragments in CRC.

## 2. Materials and Methods

### 2.1. Biological Material

A total of 155 malignant colorectal tumors and 74 adjacent non-cancerous tissue specimens were collected from patients operated for primary CRC at the University General Hospital “Attikon”. Following the surgical excision, the specimens were histologically confirmed by a pathologist and snap-frozen in liquid nitrogen. Finally, they were stored at −80 °C to preserve RNA integrity until further use.

Clinicopathological features of the individuals used in the current study included patients′ age, tumor size, tumor location, histological grade, and TNM stage. Follow-up information, including tumor recurrence and survival data, was collected.

This research study was conducted in compliance with the 1964 Declaration of Helsinki and its later amendments and was approved by the institutional Ethics Committee of the University General Hospital “Attikon”, Athens, Greece (approval number: 31; 29 January 2009). Moreover, written informed consent was obtained from all participants.

The human metastatic CRC cell line SW-620 was propagated in Leibovitz’s L-15 Medium, adjusted to contain fetal bovine serum in a final concentration of 10%, 100 kU/L penicillin and 0.1 g/L streptomycin. SW-620 cells were seeded at a concentration of 0.5 × 10^5^ cells/mL and incubated for 48 h at 37 °C, in free gas exchange with atmospheric air before trypsinization and collection for downstream applications according to American Type Culture Collection (ATCC) guidelines.

### 2.2. Clinicopathological Parameters of CRC Tumors and Biological Characteristics of Patients

The present study included a total of 155 primary malignant colorectal tumors and 74 paired non-cancerous colorectal tissue specimens from patients who were subjected to surgical resection of their primary colorectal tumor. The study cohort consisted of 82 male and 73 female patients. The median age at CRC diagnosis was 69 years ranging from 37 to 93 years. Most of the patients (107 patients, 69%) presented a primary colorectal tumor, while the rest of them (48 patients, 31%) presented rectal tumors (Table 1). Histological examination of all malignant lesions was accomplished based on the World Health Organization (WHO) guidelines and revealed that 13 of them (8.4%) were designated as well differentiated or grade I malignant neoplasms, 115 (74.2%) as moderately differentiated or grade II and 27 (17.4%) as poorly differentiated or grade III. According to the TNM classification system, 17 (11%) colorectal tumors were characterized as TNM stage I, 65 (41.9%) as TNM stage II, 56 (36.1%) as TNM stage III, and 17 (11%) as TNM stage IV. All the biological characteristics and clinicopathological features of patients and their tumors are summarized in Table 1.

Due to missing follow-up data, 13 of the 155 patients were excluded from the downstream analysis in our study. From the remaining 142 patients with complete follow-up information, a total of 17 patients had distant metastasis (M1) at the time of surgical excision of the tumor and consequently were excluded from DFS analysis. From a total of 125 patients that were included in the DFS analysis, tumor recurrence was detected in 49 individuals (39.2%) during follow-up time intervals. Similarly, regarding the OS of the 142 patients with complete follow-up data, 73 deaths (51.4%) related to CRC were recorded. The median follow-up time was 103 ± 3.7 months.

### 2.3. Extraction, Polyadenylation, and Reverse Transcription of total RNA

All tissue specimens were homogenized and dissolved using a mixture of guanidine thiocyanate and phenol in a monophase solution, which is commercially available as TRI Reagent^®^ (Molecular Research Center, Inc., Cincinnati, OH, USA). Total RNA was extracted from SW-620 cells and homogenized colorectal tissues, diluted in THE RNA Storage Solution (Life Technologies Ltd., Carlsbad, CA, USA) and stored in deep freeze −80 °C, according to manufacturer′s instructions. The concentration and purity of the RNA were assessed spectrophotometrically at 260 and 280 nm, using a BioSpec-nano Micro-volume UV-Vis Spectrophotometer (Shimadju, Kyoto, Japan). As a final step, total RNA polyadenylation and reverse transcription into first-strand cDNA synthesis using an oligo-d(T) adapter primer were performed as previously described [29]. In more details, the polyadenylation of tsRNA was carried out using recombinant E.coli poly(A) polymerase (New England Biolabs Ltd., Whitby, ON, Canada), while the oligo-dT adapter served as primer for first-strand cDNA synthesis. Moreover, the oligo-dT adapter provides a universal priming site, enabling the efficient amplification of sncRNAs. This methodology enables the selective amplification of tsRNAs, whereas mature tRNAs cannot be reverse transcribed and subsequently amplified (Appendix A).

### 2.4. Real-Time Quantitative Polymerase Chain Reaction (qPCR)

To determine the expression levels of the target tiRNA, we developed a real-time quantitative polymerase chain reaction (qPCR) methodology based on the SYBR Green chemistry, in a 7500 Fast Real Time PCR System (Applied Biosystems, Foster City, CA, USA). Specific forward primers were designed for the amplification of 5′-tiRNA-Pro^TGG^, as well as for the endogenous reference controls, namely the small nucleolar RNAs C/D box 48 and 43 (*SNORD48* and *SNORD43*, also known as *RNU48* and *RNU43*). 5′-tiRNA-Pro^TGG^ sequence derived from MINTbase v2.0 [30] with unique sequence identifier tRF-32-6978WPRLXN48Q, while published sequences of *RNU48* and *RNU43* derived from GenBank^®^ under accession numbers NR_002745.1 and NR_002439.1, respectively (Appendix A). All forward primers were used in combination with a common reverse primer, designed to anneal to the oligo-dT-adapter (5′-GCGAGCACAGAATTAATACGAC-3′). As a result, three distinct amplicons were generated.

The final reaction mixture contained 1 μL of 10-fold diluted cDNA, 5 μL KAPA™ SYBR^®^ FAST qPCR master mix (2X) (Kapa Biosystems Inc., Woburn, MA, USA), and 2 μL of primers (final concentration: 200 nM each), in a total final reaction volume of 10μL. Cycling conditions included a denaturation step at 95 °C for 3 min followed by 40 cycles of 95 °C for 3 s, for denaturation of the PCR products and 60 °C for 30 s, for primer annealing, extension and detection of the fluorescence. As a final step, melting curve analysis was carried out by heating final reaction mixtures from 60 °C to 95 °C with a heating rate of 0.3 °C/s and continuously acquiring fluorescence emission data to distinguish specific PCR products from primer-dimers or other non-specific products, which are characterized by different melting temperature (T_m_) than those of the respective amplicons.

Each reaction was performed in duplicate to ensure high reproducibility of the obtained results. Relative quantification of 5′-tiRNA-Pro^TGG^ levels in tissue specimens was carried out using the comparative C_t_ method also known as 2^−ΔΔCt^. In brief, *RNU48* and *RNU43* were used as endogenous reference controls to normalize values obtained by qPCR for the amount of RNA used in reverse transcription; the utilization of the geometric mean of multiple reference genes is considered more accurate for the normalization of qPCR data [31]. The SW-620 cell line was used as calibrator to compare data derived from all qPCR experiments; a calibrator sample is generally used to diminish the technical variation between individual qPCR runs [32]. All the prerequisites required for the implementation of the 2^−ΔΔCt^ method were checked in a validation experiment, in which C_t_ values of 5′-tiRNA-Pro^TGG^ as well as *RNU48* and *RNU43* were measured in a dilution series of SW-620 cDNA. The efficiency (*E*) of real-time PCR was calculated using the equation: *E* = −1 + 10^(−1/^^α^^)^, where α corresponds to the slope of the amplification plot (Appendix A). The normalized result of each sample was calculated as the ratio of 5′-tiRNA-Pro^TGG^ copies to the geometric mean of *RNU48 and RNU43* copies, divided by the same ratio calculated for SW-620 cells. Finally, the relative 5′-tiRNA-Pro^TGG^ expression of each sample was measured in relative quantification units (RQU).

### 2.5. Biostatistical Analysis

Data analysis was performed using the IBM SPSS Statistics 25 software (IBM Corp., Armonk, NY, USA). The levels of 5′-tiRNA-Pro^TGG^ in both malignant colorectal tissue samples and non-cancerous specimens demonstrate a non-Gaussian distribution. As a result, non-parametric tests were utilized to assess the potential significance of differences regarding 5′-tiRNA-Pro^TGG^ levels between the abovementioned cohorts. More specifically, the analysis of differences between the two groups was performed using Mann–Whitney *U* test, and/or Wilcoxon signed-rank test if paired tissue samples were available. In the same context, the analysis of differences among distinct subgroups of patients, stratified according to each clinicopathological parameter, carried out using Mann–Whitney *U* test or Kruskal–Wallis test, where applicable. Furthermore, the potential associations between 5′-tiRNA-Pro^TGG^ expression status and the most important clinicopathological parameters was performed using the Kendal′s tau b test and two-tailed χ^2^-test or Fisher′s exact test, where appropriate.

Receiver operating characteristic (ROC) analysis was carried out to evaluate the discriminatory potential of 5′-tiRNA-Pro^TGG^ in CRC. Kaplan-Meier survival analysis was conducted to evaluate the association between 5′-tiRNA-Pro^TGG^ levels and both DFS and OS of the patients. For this purpose, patients were categorized as 5′-tiRNA-Pro^TGG^ positive or negative, based on an optimal prognostic cut-off point, corresponding to the 71st percentile and equal to 0.65 RQU, which was determined by X-tile algorithm [33]. The differences between DFS as well as OS in the obtained Kaplan-Meier curves were assessed using Mantel-Cox (log-rank) test. Bootstrapped Cox regression analysis was carried out to evaluate the prognostic value of 5′-tiRNA-Pro^TGG^ levels as an estimator of the hazard ratio (HR) regarding disease recurrence and patient death, using 1000 bootstrap samples for internal validation. The bootstrap bias-corrected and accelerated (BCa) 95% confidence interval (CI) of each estimated HR along with the respective bootstrap *p* value were also calculated. Finally, multivariate Cox regression models were constructed and adjusted, based on the most important clinicopathological covariates. The level of statistical significance was set at a probability value of less than 0.050 (*p* < 0.050).

### 2.6. In-Silico tRNA-Pro^TGG^ Target Prediction and Functional Enrichment Analysis

The identification of potential 5′-tiRNA-Pro^TGG^ targets carried out using IntaRNA and RNAhybrid [34,35]. The targets, which were identified in both prediction tools, were further analyzed. To examine the potential biological role of 5′-tiRNA-Pro^TGG^ in CRC, we performed a functional enrichment and pathway analysis of 5′-tiRNA-Pro^TGG^ target genes. More specifically, target genes were imported into ShinyGO v.0.75 [36] to perform Gene Ontology (GO) function analysis as well as Kyoto Encyclopedia of Genes and Genomes (KEGG) pathway enrichment analyses, using the default settings (0.05 false discovery rate (FDR)-adjusted *p* value as threshold).

## 3. Results

### 3.1. Comparison of 5′-tiRNA-Pro^TGG^ Expression Levels between Malignant Colorectal Tumors and Adjacent Non-Cancerous Tissues

The levels of 5′-tiRNA-Pro^TGG^ in colorectal tissue specimens ranged from 0.011 to 74.73 RQU with a median of 0.25 RQU for malignant tumors and 0.42 RQU in non-cancerous tissue specimens ranged from 0.038 to 31.18 RQU (Table 2). The comparison of 5′-tiRNA-Pro^TGG^ levels between the 74 colorectal tumors and their paired adjacent non-cancerous counterparts unveiled its significant downregulation in the great majority (82.4%) of colorectal tumors (*p* < 0.001, Figure 1). Additionally, ROC analysis illustrated the ability of 5′-tiRNA-Pro^TGG^ to discriminate CRC from normal colorectal mucosa [area under curve (AUC) = 0.61, 95% CI 0.54–0.68, *p* = 0.007] (Appendix A). Subsequently, CRC patients were classified as 5′-tiRNA-Pro^TGG^ positive or negative, based on their 5′-tiRNA-Pro^TGG^ levels, using an optimal single cut-off value, as described in the “Materials and Methods” section. However, positive or negative 5′-tiRNA-Pro^TGG^ expression status was not found to be associated with major clinicopathological variables of the patients, such as tumor size, histological grade, invasion, nodal status as well as distant metastasis (Appendix A).

### 3.2. Elevated 5′-tiRNA-Pro^TGG^ Levels Are Associated with Poor DFS Independently of the Established Prognosticators

During the follow-up time, tumor recurrence occurred in 49 patients (39.2%) from a total of 125 patients who did not present distant metastasis at the time of diagnosis (M0). Using Cox univariate regression analysis (Figure 2) a nearly 2-fold increased risk of relapse was predicted for CRC patients with tumors characterized as positive, regarding the 5′-tiRNA-Pro^TGG^ expression status [HR = 1.94, Bootstrap BCa 95% CI = 1.14–3.32, bootstrap *p* = 0.024].

Prediction of DFS using Kaplan-Meier survival analysis was in accordance with the aforementioned results. In specific, Kaplan-Meier curves illustrated a significant lower DFS for CRC patients that were characterized as 5′-tiRNA-Pro^TGG^-positive compared to 5′-tiRNA-Pro^TGG^-negative tumors (*p* = 0.021; Figure 3). Most important, the incorporation of established clinical prognosticators in a multiparametric Cox regression model highlighted the independent prognostic value of tiRNA-Pro^TGG^ expression [HR = 2.08, Bootstrap BCa 95% CI = 1.05–4.95, bootstrap *p* = 0.041].

### 3.3. Elevated 5′-tiRNA-Pro^TGG^ Levels Are Associated with Poor OS of CRC Patients

Concerning the OS of CRC patients, Kaplan-Meier curves demonstrated that 5′-tiRNA-Pro^TGG^-positive patients had lower OS probability than patients with 5′-tiRNA-Pro^TGG^-negative tumors (*p* = 0.036; Figure 3). Furthermore, survival analysis within the cohort of non-metastatic CRC patients (M0), unveiled that 5′-tiRNA-Pro^TGG^—positive patients had a lower OS time interval compared to the negative ones (*p* = 0.033; Appendix A). This tendency was further enhanced by Cox univariate regression analysis results (Figure 4), which also indicated that patients with 5′-tiRNA-Pro^TGG^-positive tumors were at a higher risk of death, compared to the negative ones (HR = 1.67, 95%, Bootstrap BCa 95% CI = 1.00–2.95, bootstrap *p* = 0.042). However, when tiRNA-Pro^TGG^ expression status was included in a multiparametric Cox regression model along with all the important clinicopathological parameters failed to retain its independent prognostic power in CRC [HR = 1.49, Bootstrap BCa 95% CI = 0.82–2.61, bootstrap *p* = 0.11].

### 3.4. Overexpression of 5′-tiRNA-Pro^TGG^ Is Associated with Low DFS and OS of Patients with Rectal Carcinomas and/or Moderately Differentiated Colorectal Tumors

To further evaluate the prognostic value of 5′-tiRNA-Pro^TGG^ expression status, patients were stratified based on important clinicopathological variables, such as tumor location and tumor grade. Initially, the significance of 5′-tiRNA-Pro^TGG^ expression status was evaluated regarding tumor location. Patients whose tumor is located in rectum and overexpresses 5′-tiRNA-Pro^TGG^ demonstrated a significantly higher probability of tumor recurrence (*p* = 0.027) (Figure 5A) and inferior OS (*p* = 0.047) (Figure 5B). Subsequently, we classified CRC patients according to the histological grade of their tumor. The lack of cases corresponding to well- and poorly differentiated tumors confined our analysis only in the subgroup of patients with moderately differentiated colorectal tumors. Finally, Kaplan-Meier survival analysis demonstrated that the elevated 5′-tiRNA-Pro^TGG^ levels in patients with grade II CRC were associated with short DFS (Figure 5C) and OS (Figure 5D) time intervals.

### 3.5. GO Term and Pathway Enrichment Analysis of Predicted 5′-tiRNA-Pro^TGG^ Targets

To acquire a better overview of the potential biological role of 5′-tiRNA-Pro^TGG^ in CRC, we performed a functional enrichment and pathway analysis of 5′-tiRNA-Pro^TGG^ target genes. A total of 951 genes were predicted by both IntaRNA and RNAhybrid as putative targets of 5′-tiRNA-Pro^TGG^. GO term enrichment analysis revealed that the predicted target genes of 5′-tiRNA-Pro^TGG^ (Appendix A) are implicated in cellular communication and signaling, several developmental processes as well as cellular motility (Figure 6A). Moreover, KEGG pathway analysis unveiled a potential association of 5′-tiRNA-Pro^TGG^ target genes with key oncogenic pathways, including AMPK signaling pathway, MAPK signaling pathway, MTOR signaling pathway and PD-L1 expression and PD-1 checkpoint pathway in cancer (Figure 6B).

## 4. Discussion

The tsRNAs constitute a recently emerged group of sncRNAs, which can serve as molecular tumor markers. These fragments are not randomly generated during tRNA degradation, but through precise endonucleolytic mechanisms, which, along with their position of origin in tRNA transcripts, constitute key aspects for their classification into various subgroups [37]. Due to the cellular dynamics of tsRNAs, it is speculated that they play a key role in the epigenetic regulation of the protein-coding genes, using RNA interference (RNAi) in a similar way to other sncRNAs such as miRNAs and piRNAs [38,39]. This observation has been further enhanced by several studies, which demonstrated a close interaction between specific tsRNAs and AGO or PIWI proteins during silencing of target genes in a broad series of human cell lines and tissues [19,20,40]. Consequently, there is an increasing interest regarding their diagnostic and/or prognostic potential in solid tumors and hematological malignancies [37,41,42,43,44,45]. Focusing on CRC, high-throughput genomic studies have reported great variations of tsRNA levels between malignant and benign colorectal tumors [25,26]. One of these studies identified two distinctive signatures of tRFs, derived from the 3′-trailer sequences of precursor tRNAs, able to discriminate between colorectal adenomas and adenocarcinomas. Even though, the notable differences in tRF levels during the sequential transition of adenoma to a malignant colorectal tumor come in accordance with the hypothesis for their involvement in different phases of the disease, further investigation of this notion is required [26].

In the current work, we evaluated for the first time the clinical significance of a proline tiRNA in CRC. This molecule derived from a tRNA^Pro^ located at chromosome 11 (11q13.5), bearing TGG anticodon sequence (5′-tiRNA-Pro^TGG^). Our results indicated decreased expression of 5′-tiRNA-Pro^TGG^ in cancerous tissue specimens compared to matched adjacent colonic mucosa. On the other hand, multivariate Cox regression analysis showed that the overexpression of 5′-tiRNA-Pro^TGG^ constitutes an adverse prognostic factor predicting short-term relapse of CRC patients independently of the established prognosticators such as patients’ age, tumor size and location, histologic grade of the tumor and TNM stage. Furthermore, univariate Cox regression analysis revealed a significantly increased probability of CRC patients overexpressing 5′-tiRNA-Pro^TGG^ to succumb earlier. The bootstrapped analysis used for internal validation and strengthened the statistical significance of these results. All the findings were further confirmed by Kaplan-Meier survival analysis, which indicated that overexpression of 5′-tiRNA-Pro^TGG^ was associated with both low DFS and OS of CRC patients.

Moreover, high levels of 5′-tiRNA-Pro^TGG^ retained their adverse association with both DFS and OS within the subgroups of non-metastatic (M0) patients as well as patients with moderately differentiated colorectal tumors and rectal carcinomas. Histological grade of the disease remains a critical prognosticator of the outcome, nevertheless, intra- and inter-observer variability during tumor grading can often lead to misclassification, especially between well- and moderately differentiated tumors [46]. The impact of misclassification varies, resulting to over- or underestimation of tumor aggressiveness and consequently over- or undertreatment of the patient. Elevated expression levels of 5′-tiRNA-Pro^TGG^ in moderately differentiated tumors (G2) may be used as an ancillary tool to identify patients at increased risk of relapse, alleviating some of the intrinsic limitations of histological grading. In the same context, overexpression of 5′-tiRNA-Pro^TGG^ tumors retained its unfavorable prognostic potential in the subgroup of patients with localized rectal cancer. Although rectal carcinomas represent one-third of total CRC cases, they differ from colon cancer in terms of anatomy, embryological origin and function and for this reason they require a different treatment approach [47].

Unequivocally our results illustrate that the increased expression of 5′-tiRNA-Pro^TGG^ constitutes an unfavorable predictor of both OS and DFS in CRC patients in line with survival data presented by OncotRF database, where 5′-tiRNA-Pro^TGG^ expression levels have been correlated with adverse prognostic outcome in a wide spectrum of malignant neoplasms including breast, stomach, prostate, skin as well as head and neck cancers [28]. However, the comparison of 5′-tiRNA-Pro^TGG^ expression levels between cancerous and non-cancerous tissue samples unveils a slightly overexpression of 5′-tiRNA-Pro^TGG^ in non-cancerous tissues (Mean ± SE: 1.73 ± 0.51 vs. 1.57 ± 0.53 RQU, respectively). As a result, the deregulation of 5′-tiRNA-Pro^TGG^ expression may represents a late event in colorectal carcinogenesis and/or 5′-tiRNA-Pro^TGG^ functions as a potential double-edged sword in CRC.

Several studies highlight the implication of tsRNA in colorectal carcinogenesis [23,48,49]. An internal fragment of tRNA^ArgCCG^ targets CLDN1 and inhibits epithelial to mesenchymal transition (EMT) of colorectal cancer cells [50]. Paradoxically, the same tRNA^Arg^ derivatives produced in a DICER1-dependent mechanism can also drive progression of colorectal cancer under hypoxic conditions [51]. This incidence points towards the direction of a potential dual-role for tsRNAs in CRC. To further enhance that point, this is not the first time such an incidence is reported for a sncRNA. There are numerous examples of miRNAs, the most well-studied class of ncRNAs, acting as “double-edged sword” in cancer generally but also in CRC specifically [52,53]. These results are in accordance with our observations regarding a potential double-edged sword role for 5′-tiRNA-Pro^TGG^. Recent studies highlight the pleiotropic functions of tsRNAs derived from tRNA^Pro^ in the fine-tuning of protein synthesis by interacting directly with ribosomes as well as the regulation of gene expression by presenting with miRNA- and piRNA-like functions [54,55,56]. Computational analysis revealed several biological processes and cancer-related pathways which are potentially regulated by 5′-tiRNA-Pro^TGG^, including AMPK signaling, autophagy, MAPK signaling and MTOR signaling. The fine-tuning of these key pathways is critical to ensure the maintenance of cellular homeostasis, whereas their deregulation drives the initiation and progression of colorectal tumors [57,58,59].

This research study revealed, for the first time, a potential prognostic value of 5′-tiRNA-Pro^TGG^ expression levels in CRC, serving as versatile marker for both rectal and colon tumors. The wide spectrum of clinical information that is provided by 5′-tiRNA-Pro^TGG^ can be utilized by medical personnel to further ameliorate the therapeutic management of the disease. Moreover, we are aware that our research failed to provide strong evidence regarding the functional role of the described tiRNA. However, in this initial attempt we mainly focused on the evaluation of the prognostic significance of 5′-tiRNA-Pro^TGG^ expression levels in CRC. Future studies should intensify their efforts towards the elucidation of the biological role of tiRNAs in CRC. At the same time, additional validation of these results, using a more homogenous patient cohort in terms of CRC type could unveil more valuable information regarding the additional clinical value of 5′-tiRNA-Pro^TGG^ in specific CRC forms.

## 5. Conclusions

In summary, the present research study unveils an adverse association of the elevated 5′-tiRNA-Pro^TGG^ expression levels with the prognosis of CRC patients. This tendency was further confirmed in patients with moderately differentiated and non-metastatic (M0) colorectal tumors as well as rectal carcinomas. Our analysis highlights the independent prognostic value of 5′-tiRNA-Pro^TGG^ expression levels concerning disease recurrence, when compared to the established prognostic markers in CRC. Moreover, bioinformatics analysis revealed a potential multifaceted biological role for 5′-tiRNA-Pro^TGG^ in CRC. A more intense functional study of the tiRNAs is prerequisite to fully uncover the intrinsic networks implicated in cancer development and progression. The clinical usefulness of 5′-tiRNA-Pro^TGG^ would be further enhanced if we consider that several tiRNAs can be found in the blood of patients with a variety of human diseases [60]. It would be intriguing to determine the levels of 5′-tiRNA-Pro^TGG^ in biological fluids, such as serum, plasma and stools of CRC patients and evaluate its role as a candidate non-invasive tumor marker. Simultaneously, our study can offer new insights in the field of biomarker research for CRC, by introducing tiRNAs as promising molecular markers, which can be used synergistically in comprehensive prognostic panels.

## Figures and Tables

**Figure 1 biomedicines-10-01120-f001:**
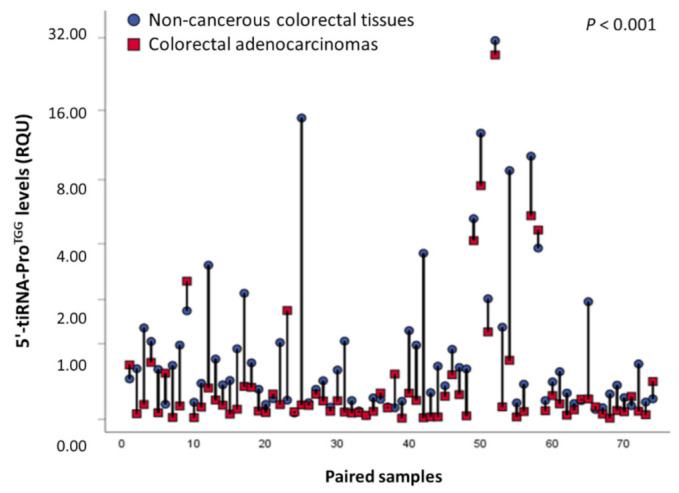
Comparison of 5′-tiRNA-Pro^TGG^ levels in pairs of malignant colorectal tissues and adjacent non-cancerous mucosae.

**Figure 2 biomedicines-10-01120-f002:**
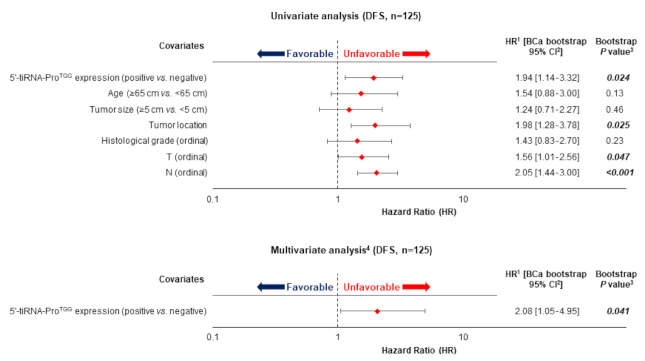
5′-tiRNA-Pro^TGG^ expression status and disease-free survival (DFS) of CRC patients. Multivariate models regarding DFS were adjusted for patients′ age, tumor size, histological grade, tumor location, depth of tumor invasion (T) and regional lymph node status (N).

**Figure 3 biomedicines-10-01120-f003:**
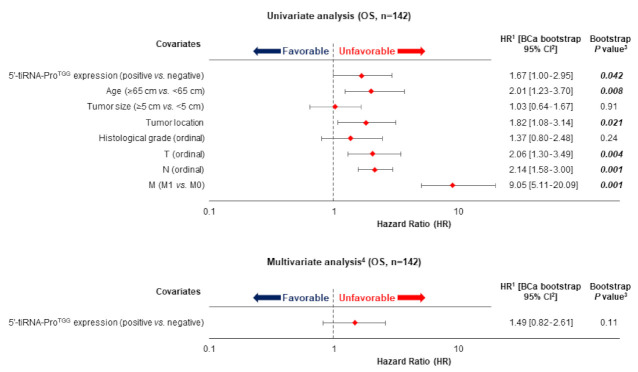
5′-tiRNA-Pro^TGG^ expression status and overall survival (OS) of CRC patients. Multivariate models regarding OS were adjusted for patients′ age, tumor size, histological grade, tumor location, depth of tumor invasion (T), regional lymph node status (N) and presence of distant metastasis (M).

**Figure 4 biomedicines-10-01120-f004:**
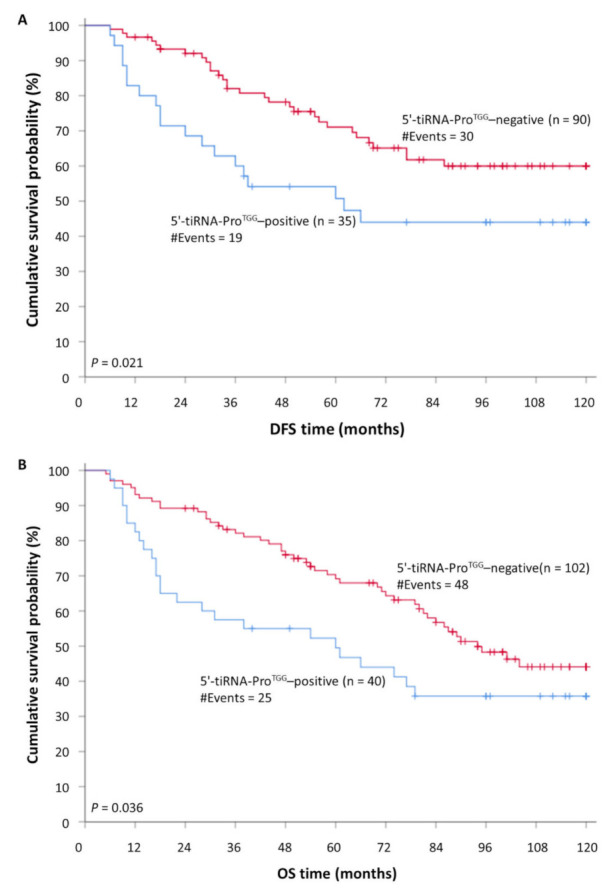
Kaplan–Meier survival curves for (**A**) disease-free survival (DFS) and (**B**) overall survival (OS) of colorectal cancer (CRC) patients. CRC patients with 5′-tiRNA-Pro^TGG^-positive tumors demonstrate significantly shorter DFS and OS intervals, compared to patients with 5′-tiRNA-Pro^TGG^-negative ones.

**Figure 5 biomedicines-10-01120-f005:**
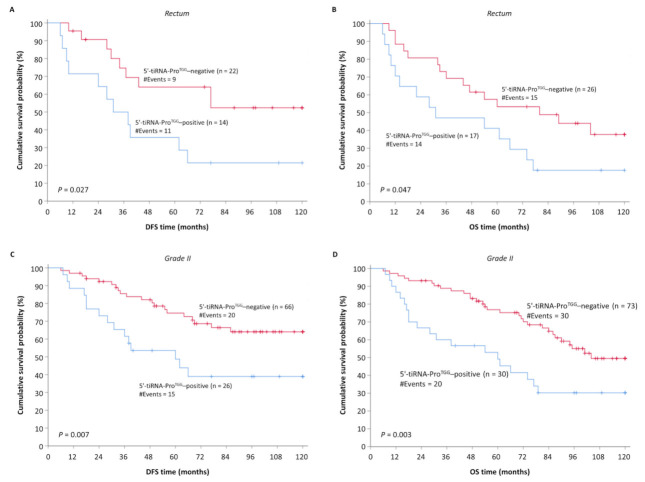
Kaplan–Meier survival curves for DFS and OS of CRC patients stratified according to location and histological grade of the primary tumor. (**A**,**B**) The prognosis of patients with rectal tumors strongly expressing 5′-tiRNA-Pro^TGG^ was significantly worse regarding both DFS and OS compared to the one of patients with 5′-tiRNA-Pro^TGG^–negative rectal tumors. (**C**,**D**) Similarly, Kaplan-Meier analysis depicted a significantly worse prognosis concerning DFS and OS in patients with moderately differentiated tumors (Grade II), overexpressing 5′-tiRNA-Pro^TGG^, compared to the ones with lower expression levels of 5′-tiRNA-Pro^TGG^.

**Figure 6 biomedicines-10-01120-f006:**
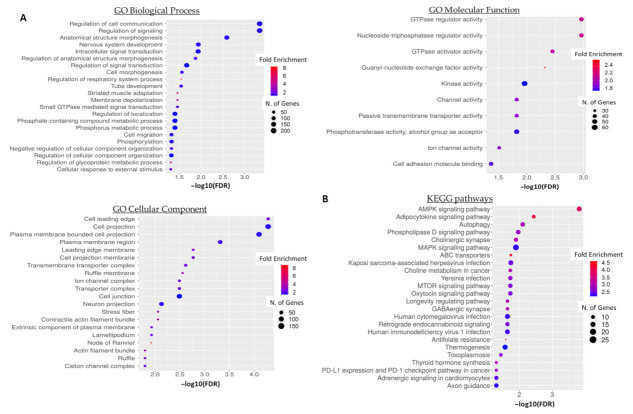
(**A**) Gene Ontology (GO) enrichment and (**B**) KEGG pathway functional enrichment analyses of the predicted 5′-tiRNA-Pro^TGG^ targets. The color and size of each bubble represent the positive fold enrichment and the number of target genes enriched in a GO term, respectively.

**Table 1 biomedicines-10-01120-t001:** Baseline clinical data.

	Number of Patients (%)
**Age**	
<65	52 (33.6%)
≥65	103 (66.4%)
**Tumor site**	
Colon	107 (69.0%)
Rectum	48 (31.0%)
**Tumor size**	
<5 cm	94 (60.6%)
≥5 cm	61 (39.4%)
**Histological grade**	
I	13 (8.4%)
II	115 (74.2%)
III	27 (17.4%)
**T (tumor invasion)**	
T1	3 (1.9%)
T2	17 (11.0%)
T3	96 (61.9%)
T4	39 (25.2%)
**N (nodal status)**	
N0	85 (54.8%)
N1	39 (25.2%)
N2	31 (20.0%)
**M (distant metastasis)**	
M0	138 (89.0%)
M1	17 (11.0%)
**TNM stage**	
I	17 (11.0%)
II	65 (41.9%)
III	56 (36.1%)
IV	17 (11.0%)
**Chemotherapy**	
Yes	78 (50.3%)
No	77 (49.7%)
**Radiotherapy**	
Yes	25 (16.1%)
No	130 (83.9%)

**Table 2 biomedicines-10-01120-t002:** Expression of 5′-tiRNA-Pro^TGG^ in 155 colorectal cancer (CRC) patients, highlighting the mean 5′-tiRNA-Pro^TGG^ levels (RQU) expression level as well as the observed expression range.

Variable	Mean ± SE	Range	Percentiles		
			25th	50th (Median)	75th
5′-tiRNA-Pro^TGG^ levels (RQU)					
in malignant tumors (*n* = 155)	1.57 ± 0.53	0.011–74.73	0.083	0.25	0.74
in non-cancerous tissues (*n* = 74)	1.73 ± 0.51	0.038–31.18	0.19	0.42	1.03

Abbreviations: RQU, relative quantification unit; SE, Standard error.

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
