# Peer review of "High Expression of a tRNAPro Derivative Associates with Poor Survival and Independently Predicts Colorectal Cancer Recurrence"

_biomedicines, 2022, doi:10.3390/biomedicines10051120_

Round 1
Reviewer 1 Report
The article is very interesting having identified an independent prognostic factor for colorectal cancer (CRC) that is the second most lethal cause of cancer-related deaths in Europe. These tRNA derivatives, in particular the analyzed tiRNAPro, play an important role in the regulation of different cellular pathways and processes. Identifying this as a marker this kind of malignant tumor could open the doors to a research field for its detection in biological fluids with minimally invasive methods and this is very important. As I said below I think other experiments for example to confirm some of the targets that have been identified by only bioinformatic analysis could been made.
Lines 37-38: In the statement I would cut the term “stepwise” to leave only “the progression”
Lines 47-51: This statement I think could be expressed in a more semplified manner. For example: “The research on predictors biomarkers to detect colorectal cancer is an emerging point of interest based on the different step of the devolpment process of the malignant tumor from a benign form”
Lines 59-61: In this statement I wouldn't use the term “tailored” but “personalized”, not the term “impending” but “upcoming” and not “intense” but “stronger”.
Line 63: In this statement I wouldn't use the term “renders” but “makes”
Lines 68-69: In this statement I only would say “Among these small ncRNA an emerging frontier is represented by microRNAs (20-22nt) that have obtained a clinical relevance only after several researchers’ efforts”
Line 84: In this statement I wouldn't use the term “diverse” but “several”
Line 162: pulverized? What do you mean?
Line 180: Are RNU48 and RNU43 used as houskeeping for the expression analysis? Sorry but I think it would be specified when you mentioned them for the first time.
Line 200: I think it would be better to do each reaction in triplicate, why did you think the duplicate would have been sufficient?
Line 242: In this statement you say that the identified target have been analyzed but this have been made only by bioinformatic analysis, is it? Why did you not think to do experiments to verify anyone of the identified target?
Author Response
Our itemized responses appear in the attached file.

Reviewer 2 Report
In this paper by Tsiakanikas et al the authors investigate the expression levels of a tRNA, 5’tiRNA-ProTGG in patients with malignant colorectal tumors, with a view to use this as a prognostic marker . They demonstrate via various analysis of the expression and patient data that high levels of 5’tiRNA-ProTGG predict both poor disease free survival and overall survival in colorectal cancer patients. The manuscript reference work highlighting that tRNA can act similarly to miRNA to regulate gene expression at translational level. They investigate this possible function for 5’tiRNA-ProTGG via a bioinformatic anaylsis of putative target genes involved in cellular homeostasis and important signalling pathways relevant to cancer- AMPK MAPK, MTOR, and PD-L1 expression.
The paper on the whole is written and referenced well and would be of interest to the readership of this journal. It is interesting that it shines a spotlight on an understudied area of cancer diagnostics, tRNA potential acting as gene modulators akin to miRNA. I have a few points that feel need addressing.
1.
for the expression data in figure 1 the levels are expressed as 5’tiRNA-ProTGG RQU reading the methods this RQU is relative quantitation unit is based on a normalisation against SNORD48 and SNORD43 levels and a calibrator cell line
“The SW620 cell line was used as calibrator to compare data derived from 206 all qPCR experiments; a calibrator sample is generally used to diminish the technical var-207 iation between individual qPCR runs [32]. All the prerequisites required for the imple-208 mentation of the 2−ΔΔC_t_ _method were checked in a validation experiment and shown to be 209 fulfilled. The relative 5'-tiRNA-ProTGG expression of each sample was measured in relative 210 quantification units (RQU). 211
I do not think this is very clear I think it would be good to include this data showing this normalisation in the supplementary figures.
2
I understand that the paper is focused on the potential for 5’tiRNA-ProTGG as a diagnostic marker for predicting disease outcome and so the manuscript does not explore the mechanisms behind 5’tiRNA-ProTGG effects.
It would however be of greater benefit for the research community if the authors are able provide data set listing the gene names for the bioinformatic analysis in figure 6 showing fold differences in key genes in the molecular pathways.
minor point
line 27 28 in abstract does not read well “ regulation of critical for cellular homeostasis”
Author Response

(The authors gave the same response as above.)

Reviewer 3 Report
The manuscript by Panagiotis Tsiakanikas et al investigates the correlation of 5’-tiRNA-ProTGG and prognosis in colorectal cancer.
The study is well written, clear and interesting.
Two things might improve the manuscript:
Can authors perform a ROC analysis of 5’-tiRNA-ProTGG ? It would show its potential use as biomarker
Are there data on the mutational status of studied tumors? the correlation with mutations would be interesting.
Author Response

(The authors gave the same response as above.)

Reviewer 4 Report
To the Authors:
This manuscript aims to investigated the prognostic utility of 5'-tiRNA-ProTGG levels in Colorectal cancer, highlighting its participation in the regulation of cellular homeostasis biological processes and signaling pathways. Colorectal cancer (CRC) is one of the most lethal causes of cancer-related deaths in Europe, so the current CRC prevention strategies are mainly focused on the early detection in order to identify novel molecular biomarkers. In addition, the novelty of this manuscript is that includes very interesting information about tRNAs, in particular 5'-tiRNA-ProTGG, as a novel prognostic marker in CRC patients. Nevertheless, and in spite of the significant amount of work performed, some important issues have to be consider.
- In Introduction section, the authors say that “These reasons prompted us to evaluate the expression levels of 5'-tiRNA-ProTGG, as a novel prognostic marker in CRC patients”. Could the authors explain why they decided to evaluate specifically this tiRNA and not others? They should explain it in the introduction section and add the references if they used other studies.
- In Materials and Methods section, the authors should add as a supplementary data a table describing all the specific primers they designed in this study.
- In the Results section, table 2 explains the expression of 5'-RNAt-ProTGG in 155 patients with colorectal cancer, but it is not clear what the mean and range signify, because they show very different values. Could the authors explain this table better?
- The Legend of Figure 1 should not include any explanation of the results. These should be separate.
- Figure 3 should be displayed before figure 4, for better understanding. In the same way, the authors should correct the figures references in the manuscript.
- The authors conclude that there is an adverse association of elevated 5'-tRNA-ProTGG expression levels with the prognosis of CRC patients. The first results show that the relative expression of 5'-RNAt-ProTGG is lower in most malignant colorectal samples compared to adjacent non-malignant tissues. In contrast, the different statistical analyses show a poor SDS and OS the higher the level of tiRNA expression. How do the authors explain this contradiction? Because they do not make any reference to it in the discussion.

Author Response

(The authors gave the same response as above.)

Round 2
Reviewer 3 Report
The authors have addressed the points raised and the manuscript is suitable for publication
Author Response
We would like to thank the Reviewer.
Reviewer 4 Report
Comments to the Author:
I commend the effort made by the authors to revise this manuscript. The authors have answered correctly all my questions and they have added more tables into the manuscript, as I suggested to them. In the same way, they have better explained the Table 2 Legend and corrected the order of tables. They have added a clearer explanation of the discrepancy of the relative expression of 5'-RNAt-ProTGG in the discussion section. For this reason, I consider that now the manuscript is more solid and it has improved considerably for its publication. However, I still have some issues that should be addressed. In my first review, I suggested the authors to include as a supplementary data a table describing all the specific primers they designed in this study. As I suggested them, authors have included a Table S1 but it is exactly the same that Table S2. They should correct this mistake.

Author Response
We would like to thank the Reviewer. However, wo do think there is a misunderstanding, as Table S1 and Table S2 were not the same; therefore, there was no mistake, in our opinion. In fact, Table S1 describes all the specific primers we designed in this study, as instructed during the previous revision of the manuscript.